# Comparison of Citrated Whole Blood to Native Whole Blood for Coagulation Testing Using the Viscoelastic Coagulation Monitor (VCM Vet™) in Horses

**DOI:** 10.3390/ani14192892

**Published:** 2024-10-08

**Authors:** Jessica R. Vokes, Amy L. Lovett, Max C. de Kantzow, Chris W. Rogers, Pamela A. Wilkins, Benjamin W. Sykes

**Affiliations:** 1School of Veterinary Science, Massey University, Palmerston North 4410, New Zealand; 2Department of Agriculture, Fisheries and Forestry, Canberra 2601, Australia; 3Department of Veterinary Clinical Medicine, University of Illinois Urbana-Champaign, Urbana, IL 61802, USA

**Keywords:** viscoelastic testing, equine, coagulopathy, point-of-care

## Abstract

**Simple Summary:**

Assessing blood clotting in critically ill horses is important for treatment, monitoring, and prognosticating various disease states. Viscoelastic coagulation testing provides more information on an animal’s blood clotting than traditional testing; however, it is prone to variability. A point-of-care viscoelastic coagulation monitoring device (VCM Vet™) has been used in the horse to assess the coagulation of fresh blood (native whole blood). The use of native whole blood requires testing within 4 min of collection, which can be limiting in many environments. This study aimed to compare the use of blood stored in an anticoagulant (citrate) for 1 and 4 h to native blood tested immediately. The results of this study show that testing of citrated blood gives different results to native blood in the horse.

**Abstract:**

Viscoelastic monitoring of horse coagulation is increasing due to its advantages over traditional coagulation testing. The use of a point-of-care viscoelastic coagulation monitor (VCM Vet™) has been validated for use in horses using native whole blood (NWB) but has not been assessed using citrated whole blood (CWB), a technique that might have advantages in practicality and precision. Blood was collected from 70 horses, tested in duplicate immediately using NWB (T0), and stored at room temperature as CWB for testing in duplicate at 1 (T1) and 4 (T4) hours after venipuncture for comparison to NWB. Of these horses, 20 were classified as clinically healthy and used to determine reference intervals for CWB at 1 and 4 h post-collection. There were clinically relevant differences in all measured viscoelastic parameters of CWB compared to NWB meaning that they cannot be used interchangeably. These differences were not consistent at T1 and T4 meaning the resting time of CWB influences the results and should be kept consistent. The use of CWB in this study also resulted in more machine errors when compared to NWB resulting in measurements that might not be interpretable.

## 1. Introduction

Medical complications associated with coagulation dysregulation can lead to clinically significant sequelae, including venous thrombosis, laminitis, and disseminated intravascular coagulation in horses [1,2]. These sequelae contribute to morbidity and mortality, making assessment of coagulation in horses an important part of critical care [1,2]. Due to advances in technology, traditional coagulation testing (e.g., prothrombin time and activated partial thromboplastin time, fibrin degradation products, thrombin-antithrombin complexes, and platelet count) is being replaced by, or supplemented with, viscoelastic testing due to its more comprehensive assessment of coagulation [3,4].

The first viscoelastic testing devices assessed coagulation by measuring impedance changes in a blood sample using an oscillating cup and fixed probe (TEG^®^) or a vibrating probe in a fixed cup (ROTEM^®^, Sonoclot^®^). These techniques utilize citrated whole blood (CWB) that is rested and recalcified immediately before testing, with the addition of a variety of activators to limit within-horse variability [5,6]. Despite these steps, variability remains [7], possibly due to preanalytical errors. To address this, standardized methodology protocols have been developed with variable success [5,8,9]. More recently, a point-of-care (POC) device (VCM Vet™, Entegrion, Durham, NC, USA) has become available to veterinarians. This POC analyzer utilizes a cartridge-based system that activates clotting of native whole blood (NWB) through contact activation between two moving parallel glass discs, thus eliminating the need for anti-coagulation, clotting activators, or sample dilution, which should, in turn, limit preanalytical variation. While the device is simple to use, variability has been demonstrated secondary to prolonged holding and increased handling of NWB, limiting the devices’ use to when a trained operator is available to perform testing [10]. Recent work has shown that coagulation testing with the VCM Vet™ can be undertaken using CWB in dogs with acceptable variability. However, clotting time (CT) and maximum clot formation (MCF) were different compared to NWB [11]. This suggests the determination of CWB reference intervals for the use of this analyzer may be appropriate in dogs, potentially increasing the utility of the device across clinical situations [11].

The main objective of this study was to assess viscoelastic testing results obtained using CWB samples, rested for 1 and 4 h before analysis, compared to NWB samples using the VCM Vet™ device. A secondary objective of this study was to determine an institutional reference interval for the horse using CWB and the VCM Vet™ device. We hypothesized that CWB held for 1 and 4 h would have reduced CT and increased MCF compared to NWB, with the remainder of parameters comparable to NWB, and that repeatability would be acceptable for the determination of institutional reference intervals, similar to results reported in dogs [11].

## 2. Materials and Methods

### 2.1. Animals

This prospective study involved both client and university-owned horses. For the comparison of CWB to NWB, horses were eligible for inclusion if they were 1 to 25 (inclusive) years old, had a jugular vein that was not used for venipuncture in the three days prior, and had no history of pro- or anti-coagulant therapy in the 14 days prior. From those eligible for inclusion, horses were selected at convenience from cases presented to the Massey University Equine Veterinary Clinic or from the Massey University equine teaching herd. The information recorded for each horse included breed, age, sex, reason for presentation (hospitalized horses), history of pro- or anti-coagulant use, any medications administered in the prior 24 h, physical examination data, body condition score (BCS; graded out of 9 using the Henneke body condition scoring system [12]), and body weight (Tru-Test™ or subjectively estimated if not measured, n = 10). The study was approved by the University Animal Ethics Committee (Protocol number 12/43).

### 2.2. Blood Sampling and Testing

Two investigators (AL and JV) performed all sampling and viscoelastic testing. Native whole blood was collected by direct jugular venipuncture using a 20 Ga, 1-inch needle and 6 mL syringe without anticoagulant. Before analysis, the needle was removed, and the syringe was inverted downwards for a few drops of blood to be discarded from the syringe before NWB was placed into 2 prewarmed test cartridges (VCM Vet™ Test Cartridge, Entegrion, Durham, NC, USA) within 4 min of collection, for T0 testing. A new needle was then attached to the blood syringe and inserted into two upright citrate tubes (non-siliconized polyethylene terephthalate tubes; BD Vacutainer^®^ Plus Citrate Tubes, Becton, Dickinson and Company; Franklin Lakes, NJ, USA), filling by vacuum (1.8 mL) one at a time as recommended by the manufacturer to obtain a citrate-to-blood ratio of 1:9 (for T1 and T4 testing). These citrate tubes were filled within 4 min of blood collection [10] and were kept at room temperature (18 °C) before adding blood and throughout the holding period. Lastly, the remaining blood was placed into an EDTA tube (2 mL; BD Vacutainer^®^ EDTA Tubes, Becton, Dickinson and Company; Franklin Lakes, NJ, USA). Once the NWB had moved into the test cartridges from the sample cups, the cups were removed, and both cartridges were loaded into two devices concurrently (VCM Vet™, Entegrion, Durham, NC, USA) for testing in duplicate. All samples in anticoagulant were gently inverted several times for mixing and then remained upright in a blood tube holder until the time of testing.

As recommended by the manufacturer, the two viscoelastic devices underwent daily calibration, and the machines and all cartridges were prewarmed to 37 °C prior to use (VCM Vet™ Heater Plate, Entegrion, Durham, NC, USA). The order of cartridge filling (machine A or B) and filling of the two citrate tubes (T1 or T4) were randomized by coin toss, and the EDTA tube was always filled and mixed last. All tests were run for 60 min, the maximum time allowed by the specifications. Reported test results of CT, clot formation time (CFT), alpha angle (AA), MCF, clot amplitude at 10 (A10) and 20 (A20) minutes after CT, and clot lysis at 30 (Li30) and 45 (Li45) min after CT were recorded for analysis. The EDTA tube was stored at room temperature before submission for complete blood count (CBC) and fibrinogen measurement at an external reference laboratory (IDEXX Reference Laboratories, Palmerston North, New Zealand) as soon as possible after collection. Recorded CBC analytes included red blood cell (RBC) count, hemoglobin concentration (Hb), hematocrit (HCT), platelet count, total leukocyte count, and differential leukocyte counts including neutrophils, lymphocytes, monocytes, eosinophils, and basophils. Fibrinogen was determined using the heat precipitation method.

For T1 testing, 1 h after commencing the testing of NWB samples, one CWB sample was recalcified by adding 680 µL CWB by pipette to a mixing vial containing 40 µL of 0.2 M calcium chloride, the vial was gently inverted for mixing, and approximately 0.3 mL recalcified CWB was transferred into each of two prewarmed cartridges for testing in duplicate as above. The ratio of calcium to CWB used in the present study was determined by recommendations in horses [13]. For T4, the process of re-calcification was repeated 4 h after venipuncture using the second CWB sample.

### 2.3. Determination of Reference Intervals

For the determination of reference intervals, animals were included from the above-described group if physical examination was normal, BCS was below 7.5 [14], no clinical illness was observed, and no systemic medications were administered before enrollment; venipuncture was performed without concerns, and erythrocyte-related (RBCs, Hb, and HCT) and fibrinogen measurements were within normal limits.

### 2.4. Statistical Analysis

A power calculation was performed with an online calculator [15] using reference intervals for CT in healthy adult horses [16], assuming an alpha of 5%, power of 80%, and an expected difference of means of ≤10%. This determined that a minimum of 40 horses were required. However, more variability than expected was observed during data collection. Enrollment was extended to 70 horses to account for this.

Statistical analyses were performed using R version 4.3.1 in R Studio [17]. Data handling and summarization were performed using the tidyverse packages [18]. All samples with machine-reported errors were excluded from the analysis. In all analyses, *p* ≤ 0.05 was considered significant.

Box plots and violin plots were produced of the results for each parameter using the ggplot2 and ggpubr packages [19]. Bland–Altman plots were produced to assess the agreement between the results at T1 and T4, compared to T0, for each parameter using the MKinfer package in R [20]. The Shapiro–Wilk test was used to assess if the differences were normally distributed. Non-parametric limits of agreement were calculated for the Bland–Altman plots using the MKinfer package [21]. The mean of technical duplicates was used for Bland–Altman analysis and figures where a result was available for both replicates.

Mixed effects linear models were used to assess the differences between the results for each parameter at each time point. The models were produced using the lme4 and lmerTest packages in R version 4.3.1 [22] with the values for each parameter as the dependent variable and the timepoint as the independent variable. Within the models, the sample and machine were included as fixed effects. The residuals were plotted to ensure that the assumption of normality was met. The intraclass correlation coefficient was calculated using the sjstats package [23].

A logistic regression model was used to assess the differences in failure rates at each timepoint. The model was produced using the glm function in R with the test outcome as the dependent variable and the timepoint as the independent variable. The model’s residuals were plotted to ensure the model’s assumptions were met.

Reference intervals (95%) for each measurement were calculated from the previously defined healthy horse group according to the ASVCP reference interval guidelines [24] using the non-parametric method in the referenceIntervals package in R version 4.3.1 [25]. Calculated reference intervals were rounded to the nearest integer to aid clinical interpretation. The coefficient of variation was calculated based on all values at each timepoint for each measured VCM Vet™ parameter.

## 3. Results

### 3.1. Animals

Seventy horses were enrolled, including 31 females, 31 geldings, and 8 colts and stallions. The median age was 8 (range 1–25) years old. A wide range of breeds and crossbred horses were included (presented in Appendix A along with the remaining signalment data). Median weight was 524 (range 80–680) kg and median BCS was 5 (range 3–9) out of 9. Horses presented primarily for lameness (n = 14, including 1 emergency presentation for septic navicular bursa), oral examination/dental correction (n = 7), skin lesions and masses (n = 8), non-emergency surgery (n = 7; 5 for castration, 1 for upper airway surgery, 1 for cystolith removal), euthanasia (n = 5; 3 for behavioral complaints, 2 for chronic lameness), ophthalmic disease (n = 4), behavioral complaints (n = 4), reproductive evaluation (n = 3), respiratory signs (n = 2), weight loss (n = 2), colic (n = 2), discharging tracts (n = 2), enterocolitis (n = 1), and gastroscopy (n = 1). The remaining horses were companions to presented cases (n = 2) or from the university herd (n = 6).

Twenty horses met the inclusion criteria for the determination of reference intervals. These included 7 females, 11 geldings, and 2 stallions. The median age was 7.5 (range 3–19) years old. Median weight was 539 (range 470–680) kg, and median BCS was 5.3 (range 3–7) out of 9. Erythrocyte-related and fibrinogen measurements were within normal limits for the laboratory (RBC 7.0–11.8 × 10^12^/L; Hb 112–180 g/L; HCT 0.31–0.52 L/L; fibrinogen 1.4–5 g/L). The reason for presentation for these horses included investigation of skin masses (n = 5), dental examination (n = 3), lameness investigation (n = 3), university-owned horses (n = 3), ophthalmic examination (n = 2), and one each of distal limb dermatitis, routine castration, euthanasia for behavioral concerns, and pregnancy diagnosis.

### 3.2. Viscoelastic Testing Results

Of the 420 viscoelastic tests performed, 38 tests reported a machine error (“possible sample drying” error) affecting 19 horses, resulting in 382 tests for analysis. Of the machine errors, 3 occurred at T0 (2.1% of T0 tests), 26 at T1 (18.5% of T1 tests), and 9 at T4 (6.4% of T4 tests). There was a higher proportion of machine errors at T1 compared to T0 (*p* < 0.001) but not at T4 compared to T0 (*p* = 0.092).

The measured variables for each time point are presented in Figure 1. There were differences between NWB (T0) and CWB (T1 and T4) for all measured parameters.

The mean difference from T0, the limits of agreement, and the estimates of the difference between T0 and T1, and T0 and T4, as determined by the mixed effects linear models, are presented in Table 1. The coefficient of variation for all samples, and 95% reference intervals for the 20 horses defined as healthy, for all parameters at all time points are presented in Table 2.

## 4. Discussion

The findings of this study demonstrate that the use of CWB with the VCM Vet™ device in horses leads to differences in all parameters compared to NWB. These findings partially support the hypotheses of shortened CT and increased MCF when using CWB but reject the hypotheses of comparable results for the remaining measurements. This study also demonstrates that the differences between CWB and NWB were not consistent between T1 and T4. This suggests that equine CWB is not stable for viscoelastic testing with the VCM Vet™ device between 1 and 4 h at room temperature and raises concerns for varied storage time affecting viscoelastic testing results.

The measured CT using CWB was reduced from NWB, consistent with our hypothesis and previous findings in dogs [11]. While the reduction in CT at T1 compared to NWB is of a small magnitude and might not be clinically relevant, the reduction seen at T4 is more substantial. Broadly, CT informs about thrombin generation and reflects coagulation factor activity [13,26,27], with a low measurement reflecting a hypercoagulable state. The use of CWB in human viscoelastic testing has demonstrated an initial period of “instability” lasting around 30 min, during which time testing is highly variable and unreliable [28,29]. Following this initial period, CWB is stable for viscoelastic testing in humans [28,29]. Due to species differences, this might not be true of equine CWB. A possible explanation for this finding is that citrate does not completely inhibit coagulation in vitro [28], and time is required to reach the stability of subsequent enzymatic activity. Citrate provides anticoagulant activity by binding calcium ions, which act as essential cofactors to activate various clotting factors. However, the activation of factor XII and subsequent activation of factor XI are not calcium-dependent, and activation of these factors represents the initial steps toward thrombin generation in the contact-activated pathway. This phenomenon has been demonstrated in horses by increased factor XII and XI activity in CWB samples stored for up to 30 min, contributing to reduced CT using a different viscoelastic testing device [30]. Initiation of coagulation via the tissue factor pathway can also contribute to reduced CT [6,31], such as that seen with traumatic venipuncture [32]. In the present study, only a small blood volume was discarded before analysis in line with manufacturer recommendations and previous studies using this device [10,16]. The rationalization of discarding blood before analysis is to limit the effect of possible coagulation activators (e.g., tissue factor) by discarding the portions of blood most likely to contain them. Discarding a volume of blood is weighed against increased sample agitation, which also activates coagulation. Recommendations have been made to discard a blood sample before filling of citrate tube used for TEG^®^ and ROTEM^®^ assays when atraumatic venipuncture is not achieved [33]. While the venipuncture technique might have affected CT in this study, utilizing the same blood sample for all time points makes this an implausible explanation for the effect seen. Filling the citrate tubes by vacuum is another consideration for these findings, as this is likely to agitate blood and could have contributed to the reduced CT observed. The decision to fill citrate tubes in this way was performed to mimic clinical practice and to more accurately obtain the desired citrate-to-blood ratio.

Additional findings of the present study were reduced AA, A10, and A20 and increased CFT of the T1 CWB sample compared to the T0 NWB sample. These effects appeared to be at least partially transient with measures at T4 more comparable to T0, although some differences remained different. These findings are inconsistent with a similar study in dogs, which showed no effect using the same device [11]. Interestingly, in humans, an opposite effect of reduced kinetic time (comparable to CFT) and increased AA has been shown when CWB was compared to NWB, an effect that was exaggerated with increased time from venipuncture [34]. These measurements are closely related and reflect the speed of clot formation after initiation, reflecting thrombin, fibrinogen, and platelet activity [27,35]. As the horses acted as their own controls and a single blood collection was used for all measurement points in this study, it is expected that absolute platelet number and fibrinogen were consistent across all testing time points for a given individual [36]. However, citrate variably influences platelet adhesion and aggregation over time [37,38], and their subsequent activation and function [28]. Platelet aggregation observed in human citrated blood appears to be consistent between 15 and 120 min, after which aggregation is reduced [38]. It is possible that an increase in platelet aggregation in CWB at T1 and subsequent reduced aggregation by T4 contributed to the findings in the present study. The effects of time and temperature on platelet aggregation have been shown in equine citrated plasma [39]; however, to the authors’ knowledge, peak aggregation time has not been determined for equine CWB samples.

The increase in MCF at T4 compared to T0 was consistent with our hypothesis. However, the lack of difference between T1 and T0 was unexpected. Maximum clot formation is a measure of clot strength mainly affected by platelets and fibrinogen [27]. Similar to the effect on CFT and AA, altered platelet activity over time in CWB is the most plausible explanation for this finding. However, given the previously discussed explanation, it is unexpected that MCF at T1 is similar to T0. Given the variability observed across the population, the findings of the present study are susceptible to type 1 and 2 errors despite the increased sample size employed during the study period, which might explain these observations.

There are also differences in Li30 and Li45 measurements at T1 and T4 compared to T0. The lysis indexes are measures of clot amplitude, expressed as a percentage of MCF, at 30 and 45 min after CT, respectively, and reflect clot lysis. The high percentages seen in the CWB samples reflect increased clot stability compared to NWB and might reduce sensitivity for the detection of abnormalities of fibrinolysis. This is relevant when assessing coagulation abnormalities associated with systemic inflammation [2,40]. However, it is worth noting that fewer results were available for Li45 using NWB (20) than for CWB timepoints (32 for T1 and 52 for T4) due to CT exceeding 15 min (900 s) in many samples and the 60 min test limit for the VCM Vet™ device. Increased clot stability might be due to the prolonged artificial condition of CWB held ex vivo as endothelial mediators such as tissue-type plasminogen activator, urokinase, and plasminogen activator inhibitors play a major role in the regulation of fibrinolysis [2,40].

The coefficients of variation in the present study are variable for the parameters measured and between groups. For the measurement of CFT, the variation is far greater for T1 and T4 compared to T0. This effect is also present for AA, MCF, A10, and A20 but to a lesser extent. These coefficients of variation represent both individual and duplicate variability. However, with horses acting as their own controls, the differences between groups are likely attributed to the use of CWB and its resting time.

Another unexpected finding in this study is the increased rate of machine errors when using CWB, especially at T1, in the VCM Vet device. The machine test cartridges contain two moisture strips to prevent sample drying during analysis; these moisture strips are sealed by individual aluminum covers to prevent desiccation during storage and prewarming and are manually removed just prior to analysis. In this study, the timing of removal of these covers was performed immediately prior to venipuncture for T0 (NWB) samples and immediately prior to recalcification of CWB for T1 and T4 samples. This timing was comparable for NWB and CWB samples and in line with manufacturer recommendations. A possible explanation for the errors seen in CWB samples is the increased handling of these samples causing increased initiation of contact activation, specifically the additional passage of blood through a needle, and mixing of the CWB samples. The higher rate of these errors using CWB compared to NWB raises concerns for the reliability of the use of CWB in this device. Financial and clinical implications of machine errors should be considered when developing protocols for coagulation monitoring using this device. A study using this device to assess coagulation of NWB from Greyhound and non-Greyhound dogs reported several missing measurements in only the Greyhound group, of which two were reported as “possible sample drying” errors [41]. In the present study, these errors affected 19 horses of 8 different breeds and 5 different half-bred crosses. Due to the low number of each breed affected, the effect of breed on the rate of machine errors could not be assessed in the present study.

The population used for this study was not restricted to healthy horses to better reflect clinical practice. Unfortunately, due to the labor requirements of testing for this study, and the requirement of a jugular vein that had not been previously used for venipuncture, the sickest horses admitted during the study period were often not included, and the resulting study population was not entirely reflective of the patients presenting to the hospital during this period.

The inclusion criteria for the determination of reference intervals led to the inclusion of 20 horses; this sample size should be considered when interpreting the resulting reference intervals. The definition of clinically healthy horses used for this inclusion criteria does not entirely rule out co-morbidities, and serum biochemical testing was not performed but could have increased the confidence in the systemic health of these horses. The signalment of horses might also have influenced the results as age and sex affect viscoelastic testing in adult humans [42]. There are no published reports directly investigating breed, weight, sex, and age (excluding foals) effects on viscoelastic testing in horses. The exclusion of horses with abnormal erythrocyte measures and fibrinogen from the reference interval group is justified by the effect of hematocrit and fibrinogen on viscoelastic testing [35,43]. Platelets also contribute to coagulation; hence, platelet count can influence viscoelastic testing results [27]. Absolute platelet count was often unable to be accurately determined in horses of this study due to clumping, which was confirmed in all cases by smear examination. Due to these logistical difficulties, platelet count was not used as an inclusion criterion for reference interval determination, which might also have influenced the results.

In comparison to previously published reference intervals for NWB using a larger number of healthy horses [16], the present study’s NWB measurements demonstrated longer CT, marginally longer CFT, slightly increased Li45, and similar AA, MCF, Li30, A10, and A20. These differences in CT, CFT, and Li45 could be due to variances in methodology (e.g., needle and syringe size which is different in the present study compared to previous reports [10,16]), sample handling (e.g., inadvertent sample agitation, distance to device), technique of investigators, and/or population. However, the sample size is likely to contribute significantly. The differences in methodology in the present paper, especially in blood collection technique (e.g., needle size and blood draw volume), might also have contributed to the initial variability resulting in the decision to increase the sample size from the initial power calculation [16].

The main limitations of the current study include the differences in sampling methodology compared to previous studies in horses using the same device [10,16], the filling of citrate tubes by vacuum, and the inclusion criteria and sample size for the determination of reference intervals. While the sampling technique is likely to have affected both NWB and CWB groups similarly, the filling of citrate tubes by vacuum is unique to the CWB group. The inclusion criteria for the determination of reference intervals might have led to the inclusion of horses with coagulation abnormalities that might have been excluded if traditional coagulation testing was performed as additional inclusion criteria, or if only research horses were included. Additionally, the testing of CWB exclusively at 1 and 4 h post-collection means that conclusions cannot be made about the results between these timepoints or beyond this period.

## 5. Conclusions

In conclusion, the present study’s findings show differences between VCM-Vet™ results when comparing NWB to CWB, and over time in CWB rested for either 1 or 4 h. This limits the practicality of using CWB as the exact resting time would need to be kept as consistent as possible (e.g., exactly 1 or 4 h after venipuncture). Furthermore, the increased rate of machine errors comes with an increased financial cost associated with test repetition, and the need to repeat patient sampling due to the retrospective nature of these errors (60 min after test initiation). These factors potentially limit the utility of the use of CWB with the VCM-Vet™ device.

Native whole blood is more likely to reflect in vivo coagulation and should be used wherever possible. If circumstances exist where a delay from sampling to testing is inevitable, institutions should aim to standardize holding conditions of CWB and develop institution-specific reference intervals.

## Figures and Tables

**Figure 1 animals-14-02892-f001:**
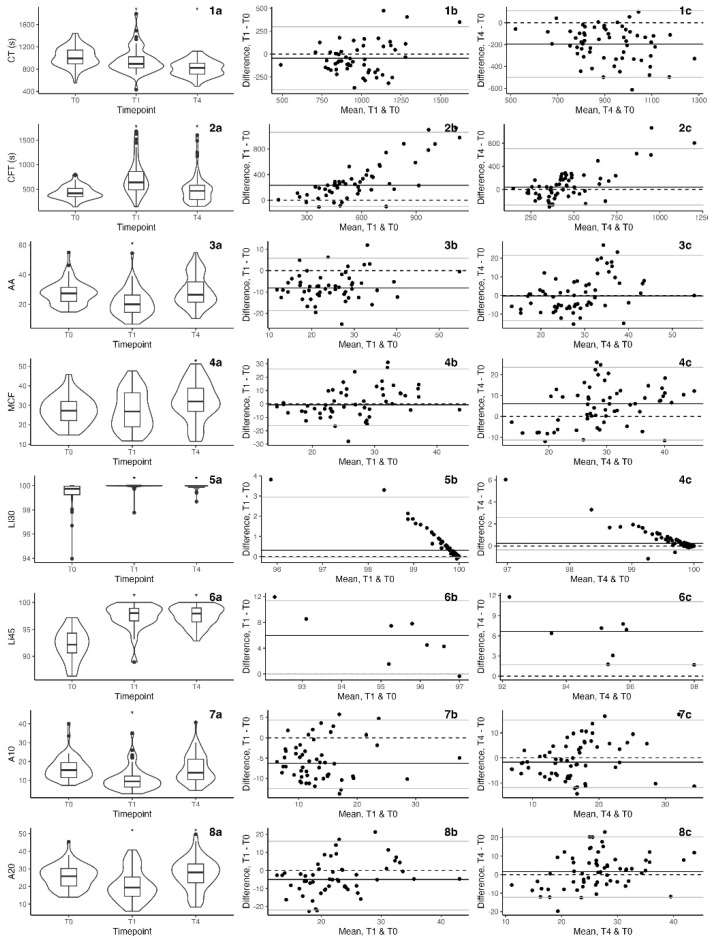
Measured values for each parameter (rows) within each time point. Data are presented in the left-hand column (**a**) as boxplots with a superimposed violin plot to illustrate density (the width of the outer area corresponds to the proportion of data at that point). Bland–Altman plots are presented in the middle and righthand columns to demonstrate the agreement between T1 and T0 (middle column; (**b**)) and between T4 and T0 (right-hand column; (**c**)). Asterisks indicate a difference between T1 or T4 and T0 based on a linear mixed model. Measurements for CT and CFT are given in seconds, AA in degrees, MCF, A10, and A20 in VCM units, and Li30 and Li45 as percentages of MCF. CT, clotting time; CFT, clot formation time; AA, alpha angle; MCF, maximum clot formation; Li30, lysis index at 30 min; Li45, lysis index at 45 min; A10, amplitude at 10 min; A20, amplitude at 20 min; T0, initial timepoint (NWB); T1, 1 h timepoint (CWB); T4, 4 h timepoint (CWB).

**Table 1 animals-14-02892-t001:** Estimates in differences in VCM Vet™ results between T0, T1, and T4 timepoints for each parameter as determined using mixed effects linear models. Estimates of bias (mean of differences) and limits of agreement from Bland–Altman analysis of the agreement between T0, T1, and T4.

VCM Vet™ Parameter	Timepoint	Estimate ± SE	*p* Value	Bias (from T0)	Limits of Agreement	Intraclass Correlation
CT	T0 (Intercept)	1027.31 ± 20.17	<0.001			0.47
	T1	−58.97 ± 16.08	<0.001	−45.05	−388.88–298.79	
	T4	−197.84 ± 15.43	<0.001	−195.01	−500.05–110.03	
CFT	T0 (Intercept)	441.02 ± 29.96	<0.001			0.39
	T1	299.4 ± 27.34	<0.001	233	−94.09–1059.53	
	T4	67.19 ± 26.25	0.011	41.25	−271.25–707.43	
AA	T0 (Intercept)	27.17 ± 1.22	<0.001			0.54
	T1	−7.13 ± 0.76	<0.001	−8.16	−18.68–5.78	
	T4	1.18 ± 0.73	0.107	−0.21	−13.47–21.59	
MCF	T0 (Intercept)	28.01 ± 1.01	<0.001			0.51
	T1	1.31 ± 0.85	0.126	−0.67	−16.13–26.02	
	T4	4.66 ± 0.82	<0.001	6.03	−11.35–23.48	
Li30	T0 (Intercept)	99.4 ± 0.07	<0.001			0.16
	T1	0.55 ± 0.08	<0.001	0.32	−0.03–2.95	
	T4	0.51 ± 0.08	<0.001	0.24	−0.36–2.59	
Li45	T0 (Intercept)	91.44 ± 0.54	<0.001			0.17
	T1	6.31 ± 0.59	<0.001	5.98	−0.01–11.34	
	T4	6.28 ± 0.56	<0.001	6.65	1.69–11.08	
A10	T0 (Intercept)	16.08 ± 0.84	<0.001			0.54
	T1	−5.64 ± 0.54	<0.001	−6.23	−12.48–4.32	
	T4	−0.32 ± 0.52	0.535	−1.7	−11.87–15.02	
A20	T0 (Intercept)	25.65 ± 0.86	<0.001			0.46
	T1	−4.69 ± 0.75	<0.001	−5.01	−21.97–16.16	
	T4	1.46 ± 0.72	0.043	1.6	−12.3–20.38	

SE, standard error; CT, clotting time; CFT, clot formation time; AA, alpha angle; MCF, maximum clot formation; Li30, lysis index at 30 min; Li45, lysis index at 45 min; A10, amplitude at 10 min; A20, amplitude at 20 min; T0, initial timepoint (NWB); T1, 1 h timepoint (CWB); T4, 4 h timepoint (CWB).

**Table 2 animals-14-02892-t002:** Calculated reference intervals for each parameter at each timepoint based on the horses defined as clinically healthy (n = 20) and calculated coefficients of variation for each parameter at each timepoint based on included tests from the entire cohort (n = 70).

VCM Vet™ Parameter	Timepoint	95% Reference Intervals	Coefficient of Variation (95% CI)
CT	T0	743–1408	0.17 (0.13–0.21)
	T1	685–1482	0.19 (0.15–0.24)
	T4	573–1104	0.17 (0.13–0.20)
CFT	T0	251–698	0.26 (0.20–0.32)
	T1	337–2494	0.52 (0.41–0.64)
	T4	197–2499	0.76 (0.60–0.93)
AA	T0	16–42	0.23 (0.18–0.28)
	T1	7–36	0.38 (0.30–0.46)
	T4	9–49	0.35 (0.28–0.43)
MCF	T0	13–42	0.22 (0.17–0.27)
	T1	8–47	0.37 (0.29–0.45)
	T4	8–40	0.30 (0.23–0.36)
Li30	T0	98–100	0.01 (0–0.01)
	T1	100–100	<0.01 (0–0.01)
	T4	100–100	<0.01 (0–0.01)
Li45	T0	88–100	0.05 (0.04–0.06)
	T1	97–100	0.02 (0.01–0.02)
	T4	93–100	0.02 (0.02–0.02)
A10	T0	8–24	0.26 (0.20–0.32)
	T1	3–17	0.43 (0.34–0.53)
	T4	4–28	0.41 (0.32–0.50)
A20	T0	13–35	0.21 (0.17–0.26)
	T1	7–40	0.43 (0.34–0.53)
	T4	6–38	0.32 (0.25–0.39)

CI, confidence interval; CT, clotting time; CFT, clot formation time; AA, alpha angle; MCF, maximum clot formation; Li30, lysis index at 30 min; Li45, lysis index at 45 min; A10, amplitude at 10 min; A20, amplitude at 20 min; T0, initial timepoint (NWB); T1, 1 h timepoint (CWB); T4, 4 h timepoint (CWB).

## Data Availability

The raw data supporting the conclusions of this article will be made available by the authors upon request.

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
