# Peer review of "Comparison of Citrated Whole Blood to Native Whole Blood for Coagulation Testing Using the Viscoelastic Coagulation Monitor (VCM Vet™) in Horses"

_animals, 2024, doi:10.3390/ani14192892_

Round 1

Reviewer 1 Report

Comments and Suggestions for Authors

A well designed methodology with some excellent points made in the Discussion.  The presentation of the results in is perhaps a bit overly complicated and the violin plots are unnecessarily redundant, but I'm happy if it is left that way.

58,59: I'm not sure that this paper concluded that producing CWD reference intervals would necessarily increase the clinical utility, but may.   

340: "no" rather than "not"

Comments on the Quality of English Language

Excellent one or two minor "typos" to correct. 

Author Response

Thank you for your comments, please see below for details regarding the changes made.   

  • 58,59: I'm not sure that this paper concluded that producing CWD reference intervals would necessarily increase the clinical utility, but may.   
    • You are correct in that the referenced paper did not conclude on the clinical utility. The wording of the sentence in the present manuscript has been changed to reflect this.
  • 340: "no" rather than "not"
    • Corrected.

Reviewer 2 Report

Comments and Suggestions for Authors

To the authors:

Thank you for your submission of this manuscript. The manuscript provides valuable information to be added to the literature as often as you pointed out horses are not in tertiary care facilities. The exploration of if citrated blood is suitable for use for VCM is an important question that needed to be answered in the literature. Overall, the manuscript is well written, and I just have a few minor points/questions.

Abstract:

OK

Introduction:

OK

Methods:

1. Did you only test T1 and T4? When evaluating some of the apTT and ACT POC devices using citrated samples the differences start to appear T2 and T3, even though clinically there would be no reason to citrate a sample and then run it, do you happen to know what T0 NWB vs T0 CWB looked like?

3. How did you determine your re-calcification protocol?

4. For the citrate tubes were they siliconized? Fine if not, I think if they are you should mention it as there is some evidence that the use of siliconized citrate tubes reduces contact activation.

Results:

OK

Discussion:

5. The CT and CFT results are widely variable, it may add to your discussion to report if this is something also seen with NWB.

Author Response

Thank you for your comments, please see below for details regarding the changes made.  

  • 1. Did you only test T1 and T4? When evaluating some of the apTT and ACT POC devices using citrated samples the differences start to appear T2 and T3, even though clinically there would be no reason to citrate a sample and then run it, do you happen to know what T0 NWB vs T0 CWB looked like?
    • Correct, we only tested CWB at 1-and 4-hours, and therefore do not have data looking at testing a CWB sample immediately after collection. We have added a statement to make this clearer to the reader (lines 380-383).
  • 3. How did you determine your re-calcification protocol
    • The ratio used for recalcification of citrated blood was as recommended by Mendez-Angulo et al. This has now been specified in the materials and methods (lines 124-125).
  • 4. For the citrate tubes were they siliconized? Fine if not, I think if they are you should mention it as there is some evidence that the use of siliconized citrate tubes reduces contact activation.
    • These were plastic, non-siliconized tubes. This specification has been added to the materials and methods section (line 93).
  • 5. The CT and CFT results are widely variable, it may add to your discussion to report if this is something also seen with NWB.
    • Thank you for raising this important point, discussion of the variability has been added to the manuscript as suggested (lines 313-318).

Reviewer 3 Report

Comments and Suggestions for Authors

 This paper investigated viscoelastic monitoring using  VCM Vet machine, testing was done with citrated whole blood (CWB) compared to native whole blood (NWB) and revealed inconsistent differences in coagulation parameters and more machine errors, suggesting CWB's lack of reliability.

The title accurately reflect the study's focus. Might be a good idea to mention the main finding but its a matter of taste.

However the abstract does not provide a clear, concise summary of the body of the paper and while  key data points are mentioned, I would suggest a clearer conclusion.  What does it mean for the reader when you mention there are machine errors and inconsistencies?

The introduction  explains the importance of comparing the new method to the standardized method and there is good clinical and practical relevance and the objectives are very clearly stated. In the M and Ms inclusion and exclusion criterias are clear (although patient selection was very heterogenous) and the sampling protocol allows for reproductibility if desired. Statistical methods are appropriate for method comparison, the rationale for each statistical approach is clear but I would have liked to see an intraclass correlation coefficient calculated as well, in order to have a better overview of consistency between the two methods but Ithink the main parameters in terms of bias and accuracy were determined. Im also missing confidence intervals for the CV.

I enjoyed the discussion as it interprets the results and explans them in contex very well. I appreciated the detailed approach to various sources of error as well but Im missing a more concise statement of limitations. I feel like an honest clear list of limitations ad potential biases will fare better and improve readability as right now the paper is difficult to read for clinicians and its hard to assess utility with so much heterogenicity in hafling, patient choice etc. I would also appreciate a simpler conclusion, more streamlined and direct. Should this sampling method be used or not. The common reader struggles for it, and Animals is adressed to a wide range of readers. What are the practical implications of using the new method, considering both strengths and weaknesses?

Author Response

Thank you for your comments, please see below for details regarding the changes made.  

  • The title accurately reflect the study's focus. Might be a good idea to mention the main finding but its a matter of taste.
    • Thank you for this suggestion; however, we feel leaving the title as subjective as possible is preferred and complies better with the journal request for a concise title. 
  • ..However the abstract does not provide a clear, concise summary of the body of the paper and while  key data points are mentioned, I would suggest a clearer conclusion.  What does it mean for the reader when you mention there are machine errors and inconsistencies?
    • We have clarified the conclusions for each of the findings mentioned in the abstract to make these clearer to the reader.
  • .. I would have liked to see an intraclass correlation coefficient calculated as well, in order to have a better overview of consistency between the two methods but I think the main parameters in terms of bias and accuracy were determined. Im also missing confidence intervals for the CV.
    • Thank you for raising these points, we have added the intraclass correlation and confidence intervals as suggested.
  • .. but Im missing a more concise statement of limitations. I feel like an honest clear list of limitations ad potential biases will fare better and improve readability as right now the paper is difficult to read for clinicians and its hard to assess utility with so much heterogenicity in hafling, patient choice etc. I would also appreciate a simpler conclusion, more streamlined and direct. Should this sampling method be used or not. The common reader struggles for it, and Animals is adressed to a wide range of readers. What are the practical implications of using the new method, considering both strengths and weaknesses?
    • Thank you for your suggestions, a paragraph has been added to the discussion to signpost the limitations of the study to the reader. The conclusion has been edited to focus on the practical implications and recommendations based on the findings of this study.